# THE MECHANISTIC BASIS OF DATA DEPENDENCE AND ABRUPT LEARNING IN AN IN-CONTEXT CLASSIFICATION TASK

**Gautam Reddy** [*]
Department of Physics, Princeton University
Physics & Informatics Laboratories, NTT Research Inc.
Center for Brain Science, Harvard University

## ABSTRACT

Transformer models exhibit *in-context* learning: the ability to accurately predict the response to a novel query based on illustrative examples in the input sequence. In-context learning contrasts with traditional *in-weights* learning of query-output relationships. What aspects of the training data distribution and architecture favor in-context *vs* in-weights learning? Recent work has shown that specific distributional properties inherent in language, such as burstiness, large dictionaries and skewed rank-frequency distributions, control the trade-off or simultaneous appearance of these two forms of learning. We first show that these results are recapitulated in a minimal attention-only network trained on a simplified dataset. In-context learning (ICL) is driven by the abrupt emergence of an induction head, which subsequently competes with in-weights learning. By identifying progress measures that precede in-context learning and targeted experiments, we construct a two-parameter model of an induction head which emulates the full data distributional dependencies displayed by the attention-based network. A phenomenological model of induction head formation traces its abrupt emergence to the sequential learning of three nested logits enabled by an intrinsic curriculum. We propose that the sharp transitions in attention-based networks arise due to a specific chain of multi-layer operations necessary to achieve ICL, which is implemented by nested nonlinearities sequentially learned during training.

## 1 INTRODUCTION

A striking feature of large language models is *in-context* learning (Brown et al., 2020; Dong et al., 2022; Garg et al., 2022; Dai et al., 2022). In-context learning (ICL) is the ability to predict the response to a query based on illustrative examples presented in the context, without any additional weight updates. This form of learning contrasts with *in-weights* learning (IWL) of query-response relationships encoded in the weights of the network. ICL emerges in transformer models (Vaswani et al., 2017) trained on a diverse set of tasks that contain a common structural element. ICL can be exploited to perform zero-shot learning on novel tasks that share this structure. For example, a transformer trained to solve numerous linear regression tasks learns to solve a new linear regression task based on in-context examples (Garg et al., 2022; Akyürek et al., 2022; Von Oswald et al., 2023; Ahn et al., 2023). Specifically, given a sequence of sample input-output pairs, the predictive error on a target query is comparable to an optimal Bayes predictor (Ahuja et al., 2023; Xie et al., 2021; Li et al., 2023). This remarkable feature extends to other generative models such as hierarchical regression models that involve model selection (Bai et al., 2023), random permutations of images (Kirsch et al., 2022) and mixture models over sequential data (Wang et al., 2023; Xie et al., 2021).

Transformer models trained on language data exhibit another simple yet powerful form of in-context learning. Given a sequence $\dots x, y, \dots, x, ?$ for $x, y$ pairs unseen during training (for example, tokens belonging to a novel proper noun), these models learn the ability to predict $y$ (Olsson et al., 2022). In other words, the model learns empirical bigram statistics on-the-fly, thus displaying a

---

[*]greddy@princeton.edu

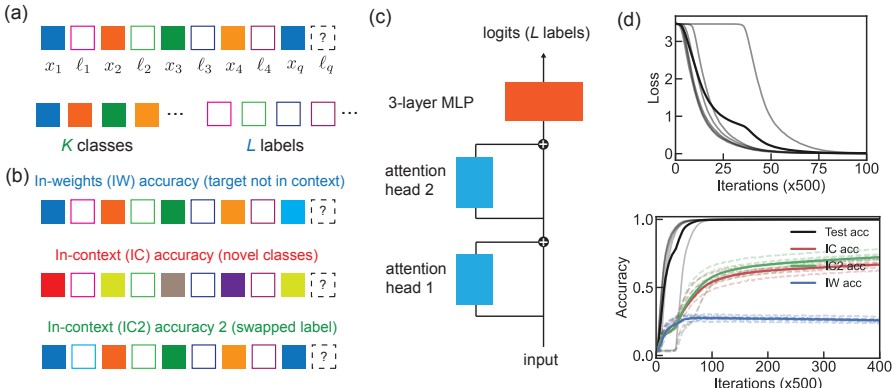

Figure 1: (a) Input sequences consist of $N$ item-label pairs followed by a target. Items are drawn from $K$ classes assigned to $L \leq K$ labels. At least one item belongs to the same class as the target. The network is tasked to predict the label of the target. The number of classes ($K$), their rank-frequency distribution ($\alpha$), within-class variability ($\varepsilon$) and the number of items from a single class in an input sequence ($B$) parameterize the data distribution. (b) IWL is measured using input sequences where the items' and target's classes are randomly sampled. ICL is measured using items and targets from novel classes and by swapping the label of an existing class in the context. (c) Network architecture. (d) Loss and accuracy curves for six seeds (dark lines show averages over the seeds). Here, $B = 2, K = 512$.

primitive form of zero-shot associative learning. Past work has shown that this computation involves an *induction head* (discussed in detail further below) and that a minimal implementation requires a two-layer attention-only network (Olsson et al., 2022; Bietti et al., 2024). Across networks of different scales and task structures, the ability to perform ICL often increases abruptly during training (Olsson et al., 2022). The mechanistic basis of the abrupt transition remains unclear. Notably, this abrupt transition is often preceded by the formation of induction heads in intermediate layers of the network, suggesting that induction head formation may provide a scaffold for the development of more complex in-context computations. Other work provides empirical evidence that ICL is the key driver behind the emergent abilities of large language models (Lu et al., 2023). Thus, elucidating the mechanisms that underpin ICL, and induction heads in particular, may provide crucial insights into the data distributional and architectural factors that lead to emergent zero-shot learning.

A recent empirical study has highlighted key data distributional properties pertinent to language that promote ICL in a hybrid in-context/in-weights classification task (Chan et al., 2022). In this setup, a 12-layer transformer network is trained to predict the class label of a target item given a sequence of $N$ item-label pairs in the context. The item classes are drawn from Omniglot (Lake et al., 2019), a standard image-label dataset. By manipulating the distribution of classes shown during training, various data distributional properties that influence the ICL vs IWL trade-off were identified. This setup offers a well-controlled paradigm for identifying the factors that enable attention-based models to learn in-context learning solutions without explicitly trained to do so.

Our main contributions are as follows. We first show that the data dependencies highlighted in Chan et al. (2022) are recapitulated in a task with simplified input statistics and a two-layer attention-only network architecture. By identifying progress measures and designing careful experiments, we show that ICL is driven by the abrupt formation of an induction head. We construct a minimal two-parameter model of an induction head stacked with a deep classifier, which reproduces all data distributional dependencies and captures the dynamics of learning. Finally, we develop a phenomenological model of an induction head's loss landscape. This analysis enables us to trace the abrupt learning phenomenon to cliffs in the landscape created by nested nonlinearities in a multi-layer attention-based network.

## 2 TASK AND NETWORK ARCHITECTURE

**Task structure.** The task structure is based on a common ICL formulation. The network is trained to predict the label of a target $x_q$ given an alternating sequence of $N$ items and $N$ labels: $x_1, \ell_1, x_2, \ell_2, \ldots, x_N, \ell_N, x_q, ?$ (Figure 1a). We embed the items and labels in $P + D$ dimensions. The first $P$ dimensions encode positional information and the latter $D$ dimensions encode content. Position is encoded by a one-hot $P$-dimensional vector (we use $P = 65$ throughout). The input sequence occupies a random window of length $2N + 1$ between 0 and $P - 1$. This choice of positional encoding biases the network to learn a translation-invariant computation.

The items are sampled from a gaussian mixture model with $K$ classes. Each class $k$ is defined by a $D$-dimensional vector $\mu_k$ whose components are sampled i.i.d from a normal distribution with mean zero and variance $1/D$. The content of item $x_i$, $\tilde{x}_i$, is given by

$$\tilde{x}_i = \frac{\mu_k + \varepsilon\eta}{\sqrt{1 + \varepsilon^2}}, \tag{1}$$

where $\eta$ is drawn from the same distribution as the $\mu_k$'s and $\varepsilon$ sets the within-class variability. The re-scaling with $\sqrt{1 + \varepsilon^2}$ ensures that $||\tilde{x}_i|| \approx 1$. Each class is assigned to one of $L$ labels ($L \leq K$). The contents of the labels are drawn prior to training from the same distribution as the $\mu_k$'s. Each label in an input sequence appears the same number of times as every other label in that sequence.

Importantly, at least one item in the context belongs to the target's class. The network is trained to classify the target $x_q$ into one of the $L$ labels using a cross-entropy loss. The network can thus achieve zero loss by either learning to classify targets from the $K$ classes as in a standard in-weights classification task (IWL), or by learning a more general in-context solution (ICL).

**Parameterizing the data distribution.** The data distribution is modulated by tuning various parameters in addition to $K$ and $\varepsilon$. The burstiness $B$ is the number of occurrences of items from a particular class in an input sequence ($N$ is a multiple of $B$). $p_B$ is the fraction of bursty sequences. Specifically, the burstiness is $B$ for a fraction $p_B$ of the training data. The classes (including the target) are sampled i.i.d for the remaining fraction $1 - p_B$. The rank-frequency distribution over the classes is $f(k) \sim k^{-\alpha}$. We use $L = 32, N = 8, D = 63, \varepsilon = 0.1, \alpha = 0$ unless otherwise specified.

**Metrics for tracking in-context and in-weights learning.** To track IWL, we measure the prediction accuracy on input sequences. The target and item classes are sampled independently from the rank-frequency distribution used during training (Figure 1b). Since $K \gg N$ in our experiments, it is unlikely that the target's class appears in the context. The network therefore has to rely on IWL to correctly predict the target's class label.

The primary metric for tracking ICL is the prediction accuracy on input sequences. The target and items belong to novel classes (the $\mu_k$'s are drawn anew). The novel classes are randomly assigned one of the existing $L$ labels (Figure 1b). $B$ copies of the target (within variability $\varepsilon$) are included in the context. Since the classes are novel, the network has to rely on ICL for accurate prediction. We introduce a secondary metric for tracking ICL using input sequences where the items' labels are different from those presented during training. We measure the accuracy of the network on predicting the *swapped* label. That is, the network has to rely on ICL rather than IWL.

**Network architecture.** The inputs are passed through a two-layer attention-only network followed by a classifier. Each attention layer has one attention head with a causal mask. Given a sequence of inputs $u_1, u_2, \ldots, u_n$, the outputs of the first ($v_i$) and second ($w_i$) layers are

$$v_i = u_i + V_1 \sum_{j \leq i} p_{ij}^{(1)} u_j, \quad w_i = v_i + V_2 \sum_{j \leq i} p_{ij}^{(2)} v_j \tag{2}$$

where

$$p_{ij}^{(\mu)} = \frac{e^{(K_\mu u_j)^T (Q_\mu u_i)}}{\sum_{k \leq i} e^{(K_\mu u_k)^T (Q_\mu u_i)}} \tag{3}$$

is the attention paid by query $i$ on key $j$ in the $\mu$th layer. $Q_\mu, K_\mu, V_\mu$ are the query, key and value matrices, respectively. The classifier receives $w_n$ as input.

The classifier is a three-layer MLP with ReLU activations and a softmax layer which predicts the probabilities of the $L$ labels. We use a deep classifier to ensure perfect IWL is feasible. At least three

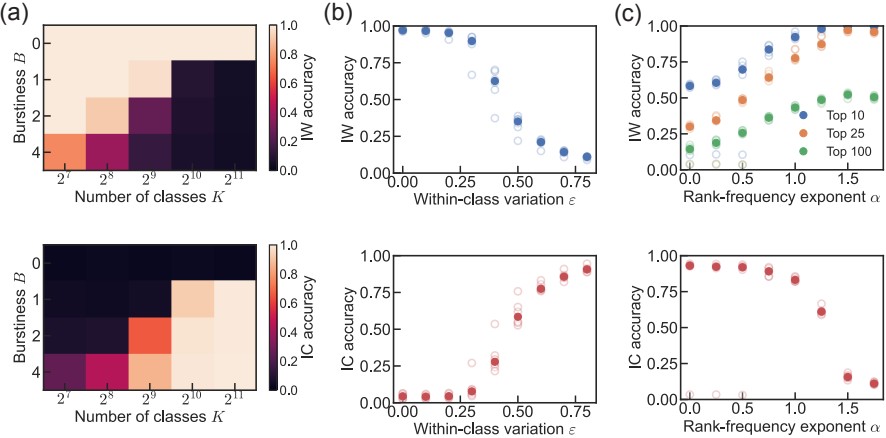

Figure 2: In-weights (top row) and in-context accuracy (bottom row) against the number of classes ($K$), burstiness ($B$), within-class variability ($\varepsilon$) and the exponent of the rank-frequency distribution ($\alpha$). Here $K = 1024, \alpha = 0, B = 1, \varepsilon = 0.1$ except when that parameter is varied.

layers were necessary to achieve perfect classification accuracy for the parameter ranges considered in this paper (since $K \gg L$). The query/key dimension and the MLP hidden layer dimension are both 128. We repeat every experiment with six seeds (with random initializations and training/test sets). For training, we use a batch size of 128 and vanilla SGD with learning rate 0.01. Figure 1d shows sample loss and accuracy curves, including the measures used to track IWL and ICL.

## 3 RESULTS

**Recapitulating data distributional dependencies.** In Figure 2, we quantify how IWL and ICL depend on the parameters of the data distribution. The upshot is that the highly simplified input statistics and network architecture considered here reproduce the core distributional dependencies observed in past work. The results are summarized below.

Increasing the burstiness $B$ and the number of classes $K$ promotes ICL while decreasing IWL (Figure 2a), highlighting the trade-off between ICL and IWL. Recall that the target and item classes are randomly sampled when $B = 0$. This implies that the network can indeed learn a perfect IWL solution for the corresponding $K$. Similarly, within-class variation ($\varepsilon$) promotes ICL and decreases IWL (Figure 2b). We find that the network always converges to an IWL solution when the fraction of bursty sequences $p_B < 1$. This is expected as the ICL solution is not a global minimum when $p_B < 1$.

A striking result is that a Zipfian rank-frequency distribution ($\alpha = 1$) overcomes the trade-off between IWL and ICL, and promotes both forms of learning. This is recapitulated in our experiments (Figure 2c). Note, however, that while the network learns the IWL solution for the most common classes, it does not learn the less frequent classes even for $\alpha = 1$.

Moreover, we find that the network can support both ICL and IWL simultaneously. To show this, we train the network on IC sequences, where the items are all drawn from novel classes randomly assigned to one of the $L$ labels. The parameter $p_C$ is the fraction of the training data containing IC sequences. The remaining fraction of the training data is drawn as described previously. When $0 < p_C < 1$ and $0 \le p_B < 1$, the network can only achieve zero loss if it learns both the in-context and in-weights solutions. Figure A.1 shows that the network is capable of learning both solutions simultaneously.

One potential explanation for the results in Figure 2 and Figure A.1 is that the network *independently* learns the in-weights and in-context solutions at different rates until it achieves zero loss. The relative rates at which the network achieves ICL and IWL will then determine the fraction of loss explained by each mechanism after convergence. The rates of ICL and IWL depend on $K, \varepsilon, B$. Specifically,

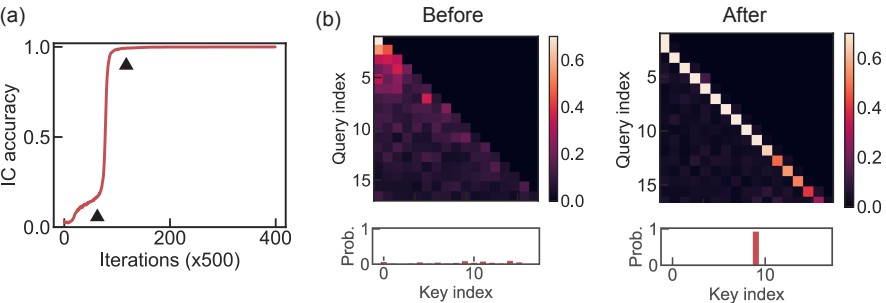

Figure 3: (a) IC accuracy curve ($p_C = 0.8, B = 1, K = 256$) shows a slow learning phase followed by the abrupt transition to zero loss. (b) The layer 1 and 2 attention maps $p^{(1)}$ (top matrices) and $p_{q.}^{(2)}$ (bottom vectors) before and after the abrupt transition (marked in the IC curve in panel (a)).

increasing $K, \varepsilon$ decreases the rate of IWL (as the classification task is harder) whereas increasing $B$ increases the rate of ICL (as there are more demonstrations in the context). The Zipfian case of $\alpha = 1$ further highlights the dynamic balance between ICL and IWL. Frequent occurrences of common classes allow the network to learn to classify them using IWL. On the other hand, the large number of rare classes promotes learning of a more general in-context solution. Once the in-context solution is learned, IWL freezes as the network incurs near-zero loss on all classes. When $\alpha > 1$, the tail of the rank-frequency distribution falls off rapidly and the rare classes do not contribute sufficiently to the loss to promote ICL. Conversely, when $\alpha < 1$, the network learns the in-context mechanism if $K$ is large enough such that IWL takes longer than ICL (see Figure 2a for $\alpha = 0$ and varying $K$).

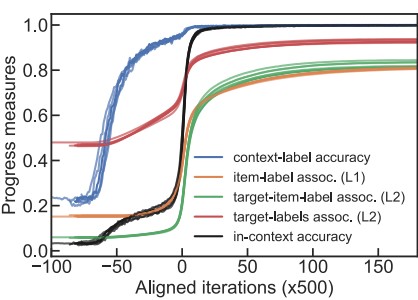

Figure 4: Progress measures for six seeds aligned based on when the IC accuracy crosses 50%. The color-progress measure pairings are orange: (ILA1), green: (TILA2), blue: (CLA), red: (TLA2), black: IC accuracy. See text for more details.

**Attention maps and progress measures.** We now examine the dynamics of ICL. We set $p_C > 0, p_B = 1$ as the IC sequences promote rapid convergence to the in-context solution and allow for more experiments. Figure 3a shows the IC accuracy, which displays a slow learning phase followed by an abrupt transition to perfect accuracy. To investigate network behavior at the transition, we examine the attention maps (for a randomly chosen input sequence) before and after the transition (Figure 3b). Before the transition, the attention map of the first layer $p^{(1)}$ shows queries paying uniform attention to the keys. For the second layer, we visualize the attention paid by the target $p_{q.}^{(2)}$ on the other tokens (as the other attention patterns do not influence classifier output), which also shows no clear pattern. After the transition, however, the attention heads show clear structure: queries in the first layer pay attention to keys that immediately precede them and the target pays attention to one particular key (here, the target's correct label).

Another curious feature of the IC accuracy curves is the slow learning phase that precedes the abrupt transition (Figure 3a). This phase leads to a non-negligible increase in IC accuracy despite the unstructured attention maps. What drives this slow learning? We hypothesize that the network learns to extract useful information from the context despite not learning the optimal ICL solution. Specifically, the total number of labels ($L$) is larger than the number of labels represented in the context ($N$). The network can thus randomly pick one of the $N$ contextual labels to increase its accuracy from $1/L$ to $1/N$. This picture suggests that the target pays attention to the $N$ labels in the second layer.

To test this hypothesis and quantify the patterns visualized in the attention maps, we define four progress measures. Item-label association (ILA1): the attention paid by a token to its previous one in the first layer. Target-item-label association (TILA2): the attention paid by the target to the correct label in the second layer. Context-label accuracy (CLA): the probability that the network predicts a label present in the context. Target-labels association (TLA2): the total attention paid by the target to the $N$ labels in the second layer. (ILA1) and (TILA2) quantify the changes that occur during the abrupt transition whereas (CLA) and (TLA2) quantify the changes expected during the slow learning phase. Each progress measure is obtained by averaging over 1000 test input sequences.

Figure 4 shows aligned progress measures (based on when IC accuracy reaches 50%). The dynamics of IC accuracy and the progress measures are remarkably reproducible across seeds. Figure 4 confirms the hypothesis that the network learns to randomly pick a contextual label in the slow learning phase (blue curve in Figure 4). Moreover, this is accompanied by the target paying attention to the labels (red curve in Figure 4). As visualized in Figure 3b, the item-label associations of the first layer and target-item-label associations of the second layer appear precisely at the transition (green and orange curves in Figure 4).

**Induction head formation drives the abrupt transition during ICL.** The dynamics of the progress measures raises various hypotheses regarding the factors that lead to ICL. Specifically, we are interested in whether learning (CLA) or (TLA2) is *necessary* for the abrupt transition (tracked by (ILA1),(TILA2)). We consider various hypotheses and design experiments to test them: H1. (CLA) $\rightarrow$ (TLA2) $\rightarrow$ (ILA1), (TILA2). H2. (TLA2) $\rightarrow$ (ILA1), (TILA2). H3. (CLA) $\rightarrow$ (ILA1), (TILA2). It is also possible that none of these factors or a progress measure that we have no considered leads to ICL.

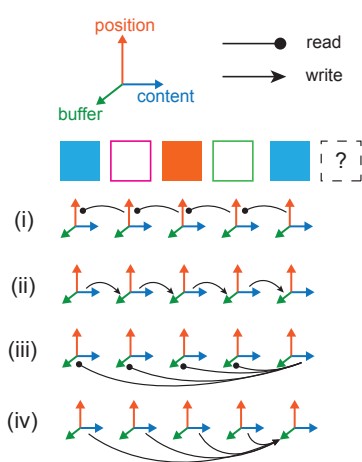

Figure 5: An illustration of the four operations performed by an induction head.

We first note that progress measures (ILA1) and (TILA2) strongly suggest the formation of an induction head. Recall that an induction head enables zero-shot copying: given an input sequence $\dots x, \ell \dots x, ?$, an induction head allows for predicting $\ell$ even if $x, \ell$ never appear together during training. Clearly, this is a mechanism that plausibly solves our task in-context. An induction head implemented by a two-layer attention-only network executes the following sequence of operations (visualized in Figure 5): (i) A token (say, $\ell$) pays attention to the token immediately preceding it (here, $x$) using positional information. (ii) The value matrix of the first layer now writes the *content* of $x$ into $\ell$. Importantly, this is written to a "buffer" subspace orthogonal to the content of $\ell$. (iii) The *target* $x$ pays attention to $\ell$ by matching its content to $\ell$'s buffer, which now contains the content of the *contextual* $x$ that preceded it. (iv) The value matrix of the second layer writes the content of $\ell$ to the target $x$, which is then passed to the classifier. The classifier in turn uses this information to predict $\ell$.

We construct a minimal three-parameter model of the two-layer induction head that emulates these core computations and also captures the four progress measures. We assume that the input embedding space can be decomposed into two orthogonal $D$-dimensional subspaces. For a token $u_i$, these orthogonal subspaces encode content $u_i^{(c)}$ and a buffer $u_i^{(b)}$ (initially empty). Given a sequence $u_1, u_2, \dots, u_n$, the first and second layers of our minimal model compute

$$v_i^{(b)} = \sum_{j \leq i} q_{ij}^{(1)} u_j^{(c)}, \quad v_i^{(c)} = u_i^{(c)} \tag{4}$$

$$w_i^{(b)} = \sum_{j \leq i} q_{ij}^{(2)} v_j^{(c)}, \quad w_i^{(c)} = v_i^{(c)} \tag{5}$$

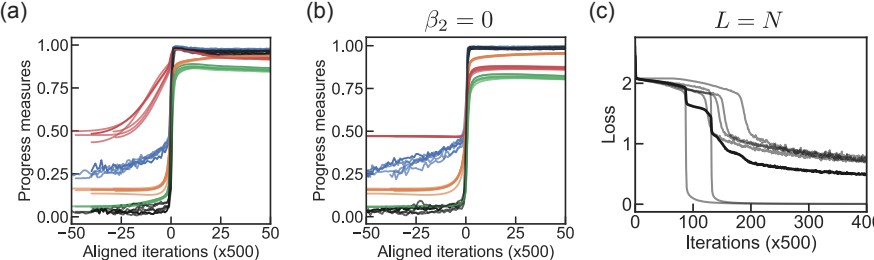

Figure 6: (a) Aligned progress measures (plotted as in Figure 4) for the minimal three-parameter model show similar dynamics as the progress measures for the full network. For $L = 32, N = 8$. (b) As in panel (a) with $\beta_2$ fixed to 0. (c) Loss curves for six seeds when $L = N = 8$.

where

$$q_{ij}^{(1)} = \frac{e^{\beta_1 \delta_{i-1,j}}}{\sum_{k \leq i} e^{\beta_1 \delta_{i-1,k}}}, \quad q_{ij}^{(2)} = \frac{e^{\alpha v_j^{(b)} \cdot v_i^{(c)} + \beta_2 \Delta_{i,j}}}{\sum_{k \leq i} e^{\alpha v_k^{(b)} \cdot v_i^{(c)} + \beta_2 \Delta_{i,k}}}. \tag{6}$$

The classifier receives the concatenated vector $w_n^{(c)} \oplus w_n^{(b)}$. Here, $\delta_{i,j}$ is one only if $i = j$ and zero otherwise. $\beta_1$ thus determines the attention paid by a token to its previous token (progress measure (ILA1)). $\alpha$ determines the attention paid by the target's content to a token's buffer (progress measure (TILA2)). $\Delta_{i,j}$ is one only if $i - j$ is odd and zero otherwise. $\beta_2$ thus determines the attention paid by the target to the labels in the context (progress measure (TLA2)). Since the classifier receives the target's content and buffer, it has the capacity to capture progress measure (CLA). We optimize for $\alpha, \beta_1, \beta_2$ and the classifier's parameters using the same training procedure as the full network. Loss and accuracy curves are presented in Figure A.2.

Progress measures from the minimal model exhibit strikingly similar dynamics (Figure 6a), including the abrupt transition in IC accuracy. Note that the slow learning phase in the IC accuracy curve is truncated in Figure 6a compared to Figure 4. Nevertheless, the network does indeed gradually learn to predict the $N$ contextual labels (blue curve in Figure 6a). The abrupt transition appears sooner for the three-parameter model, which masks the slow learning phase.

Next, we repeat the experiment fixing $\beta_2 = 0$. In this case, the target cannot pay more attention to the $N$ contextual labels relative to the items in the second layer. The dynamics of (ILA1), (TILA2) remain the same (Figure 6b), including the abrupt transition. This experiment rules out hypotheses H1 and H2, i.e., that the target-labels association (TLA2) leads to (ILA1), (TILA2).

The two-parameter model (with $\beta_2 = 0$ in equation 6) together with the deep classifier recapitulate all the data distributional dependencies exhibited by the full network (Figure A.3). Moreover, note that the two-parameter model contains only the two parameters that characterize an induction head. This reduction strongly suggests that induction head formation drives the abrupt transition in ICL in the full model.

To test hypothesis H1 that (CLA) leads to (ILA1), (TILA2), we ablate the slow learning phase. Recall that during the slow learning phase, the network learns to randomly pick one of the $N$ contextual labels. Since $L > N$, this simple strategy increases accuracy from $1/L$ to $1/N$. The slow learning phase can be prevented by setting $L = N$ and $B = 1$. That is, the input sequence contains all the $L$ labels exactly once. This perturbation indeed affects robust ICL. Specifically, two of the six seeds acquire the IC solution. The other four of the six seeds exhibit distinct, slow dynamics and converge to a sub-optimal minimum (Figure 6c).

**The loss landscape of the induction head.** We now examine the loss landscape of the induction head. Through this analysis, we aim to provide mechanistic insight into the abrupt transition and explains the empirical results described above. We propose a phenomenological model, which contains the key elements of the two-parameter induction head and the classifier. While this phenomenological approach helps identify core features of the learning dynamics, it ignores other elements. These other factors include the effects of stochasticity and the finite dimension ($D$) of the embedding. We assume $B = 1$; it is straightforward to extend the model to $B > 1$.

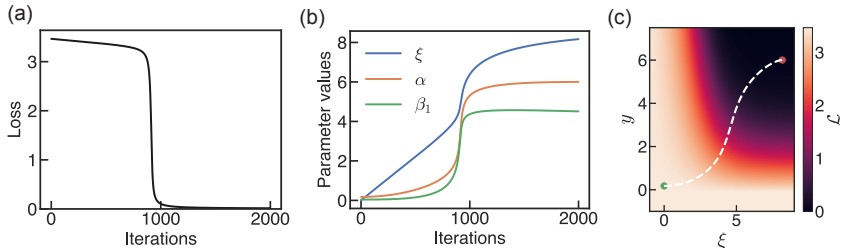

Figure 7: (a) Loss curve for the phenomenological model obtained via gradient descent on the loss in equation 9. (b) The three parameters $\beta_1$ (layer 1), $\alpha$ (layer 2), $\xi$ (layer 3) are learned sequentially. (c) The trajectory visualized on the loss landscape (green: initial point, red: final point).

Consider a softmax classifier that receives an input $w$ and classifies it into $L$ labels. Given that the target's correct label for a particular input sequence is $t$, the classifier models the probability that the label is $t$ as

$$\pi_t = \frac{e^{\gamma_t . w}}{\sum_{j=1}^{L} e^{\gamma_j . w}}, \tag{7}$$

where $\gamma_j$ is the $D$-dimensional regression vector for label $j$. The cross-entropy loss given target label $t$ is $\mathcal{L} = -\log \pi_t$. The input $w$ is given by

$$w = \frac{e^y}{e^y + 2N}\ell_\tau + \frac{1}{e^y + 2N}\sum_{k=1,k\neq\tau}^{N} \ell_k, \quad y = \alpha\frac{e^{\beta_1}}{e^{\beta_1} + N_1}, \tag{8}$$

where $\ell_j$ is the $D$-dimensional embedding vector for the label at index $j$, $\tau$ is the index of the target label $t$ in the input sequence and $N_1 = N$ for reasons discussed below. In equation 8, $y$ determines the attention paid by the target to the correct label in the second layer (recall that there are $2N + 1$ tokens in the input sequence including the target). Note that we have ignored the contributions to $w$ from the $N$ item vectors, which contain irrelevant information and add noise to $w$.

From equation 6, $y$ is the product of $\alpha$ and $v_\tau^{(b)}.v_q^{(c)}$, where $q$ is the target's index. $v_\tau^{(b)}.v_q^{(c)}$ is 1 if the label at $\tau$ pays attention to the item before it in the first layer. The attention weight corresponding to this term is $\frac{e^{\beta_1}}{e^{\beta_1}+N_1}$ (equation 6), where $N_1$ is the number of other tokens that compete for the label's attention, namely, $2\tau - 1$. Since $\tau$ varies from 1 to $N$ across input sequences, we use an intermediate value, $N_1 = N$, for simplicity. A more elaborate model would consider an expectation over the $N$ possibilities.

From equation 8, the exponents in equation 7 contain dot products of the form $\gamma_i.\ell_j$ for arbitrary pairs $i, j$. If all labels are statistically identical and balanced, it is simpler to track the overlaps $\gamma_i.\ell_i \equiv \zeta$ for all $i$ and $\gamma_i.\ell_j \equiv \zeta'$ for all $i \neq j$.

In summary, the loss after re-arranging terms is given by

$$\mathcal{L} = \log\left(1 + (N-1)e^{-z} + (L-N)e^{-z'}\right), \quad \text{where}$$

$$z = \left(\frac{e^y - 1}{e^y + 2N}\right)\xi, \quad z' = \left(\frac{e^y}{e^y + 2N}\right)\xi, \quad y = \alpha\frac{e^{\beta_1}}{e^{\beta_1} + N}, \tag{9}$$

where $\xi = \zeta - \zeta'$. The loss contains three nested logits parameterized by $\beta_1, \alpha, \xi$, which correspond to the first attention layer, the second attention layer and the third softmax layer, respectively.

The learning curves generated by gradient descent on this landscape beginning from the initial point $\xi, \alpha, \beta_1 = 0$ recapitulate the slow learning phase and the abrupt transition (for $L > N$, Figure 7a). Indeed, $\partial\mathcal{L}/\partial\xi = -(L - N)/(L(2N + 1))$ at the origin. Intuitively, when $L > N$, the classifier gradually aligns the regression vectors with the labels (increasing $\xi$) when learning to randomly pick one of the labels in the context. This phase is slow as the classifier cannot discriminate between the $N$ contextual labels. The gradual rise in $\xi$ eventually drives the loss off a cliff and leads to rapid learning of $\alpha$ and $\beta_1$ (Figures 7b,c).

As shown in Figure 6c, when $L = N$, the slow learning phase is ablated and the learning dynamics show two distinct behaviors: ICL and slow convergence to a sub-optimal minimum. We reproduce these two distinct behaviors by setting $L = N$ in equation 9 and simulating gradient descent from two points near the origin (Figure A.4a). Examining the loss landscape shows that this divergence is due to a saddle point at the origin (Figure A.4b). One path leads to the ICL solution whereas the other path gradually converges to a sub-optimal minimum. Moreover, the ICL solution takes much longer to acquire due to a shallower gradient at the origin (compare Figure 7a and A.4a). We tested this prediction in the full model (equation 2). Consistent with theory, the full model robustly learns an ICL solution for $L > N$ but not when $L = N$ (Figure A.5).

## 4 DISCUSSION

**Summary.** In summary, past work has found that particular features of the data distribution influence the trade-off between ICL and IWL. The features that promote ICL are especially prominent in language, such as a large number of rare tokens that are highly over-represented in specific contexts. We reproduced these data distributional dependencies in a simplified model, highlighting the essential ingredients necessary to explain those observations. We present strong evidence that ICL is implemented by an induction head. We build a minimal version of an induction head, which through careful experiments reveal the key factors that lead to its emergence. In particular, the learning of an independent sub-optimal strategy accompanied by a slow learning phase supports the induction head's abrupt formation. A phenomenological model of the loss landscape shows that this abrupt transition is likely due to the sequential learning of three nested logits.

**Abrupt transitions during ICL.** An abrupt transition in loss has been noted in a wide variety of ICL tasks. However, a mechanistic understanding of ICL dynamics has been lacking. Known mechanisms for ICL, such as an induction head, rely on a series of specific operations performed by multiple attention-based layers. The attention operation involves a logit (or, in general, other non-linear operations), which creates sharp gradients. A chain of operations across attention layers will thus entail a series of nested logits, which together create "cliffs" in the loss landscape. Parallel work suggests that similar mechanisms underlie 'eureka' moments in other multi-step tasks involving transformers Hoffmann et al. (2023). Note that this mechanism is distinct from the one that leads to sharp transitions in simple reinforcement learning tasks Reddy (2022).

**Relationship with past work.** Our work adds to existing evidence that induction heads play a key role during ICL Olsson et al. (2022); Bietti et al. (2024). We also recapitulate the data distributional dependencies delineated in Chan et al. (2022). Our results show that even simple networks such as ours are capable of simultaneously learning ICL and IWL solutions (see Figure A.1 for example). However, ICL is not transient in our simulations. This contrasts with recent work Singh et al. (2023) who use a much larger transformer network (12 layers and 8 heads) and finite training data. It is possible that larger networks slowly memorize the training data, leading to a gradual degradation of ICL.

**Implications for LLMs.** We show that an intrinsic curriculum may be necessary to overcome the shallow gradients at the top of the cliff and guide networks towards the ICL solution. This observation is consistent with empirical results in Garg et al. (2022), who use curricula to robustly train transformers to solve complex ICL tasks. An intriguing possibility is that learning of simpler ICL operations enables the learning of more complex ICL strategies in LLMs. An hierarchy of increasingly complex sub-tasks may lead to a cascading effect and potentially explain the sudden emergence of zero-shot learning abilities in LLMs. Testing this hypothesis will require careful mechanistic analysis of minimal networks that solve complex ICL tasks. More generally, while automatic curriculum learning has been used to train foundational models for RL Team et al. (2023), the role of curricula for accelerating ICL in LLMs remains relatively unexplored.

**Limitations.** While our formulation provides a minimal model that exhibits ICL, it is possible that larger models use different mechanisms than the ones that we have identified here. Methods for mechanistic interpretability Wang et al. (2022) may help probe these mechanisms in LLMs. We have not used heuristics such as weight tying Inan et al. (2016); Press & Wolf (2016), which are used to accelerate training of LLMs. Such heuristics may make the slow learning phase unnecessary by aligning the classifier's regression vectors with the labels (increasing $\xi$) from the outset.

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

# A   APPENDIX

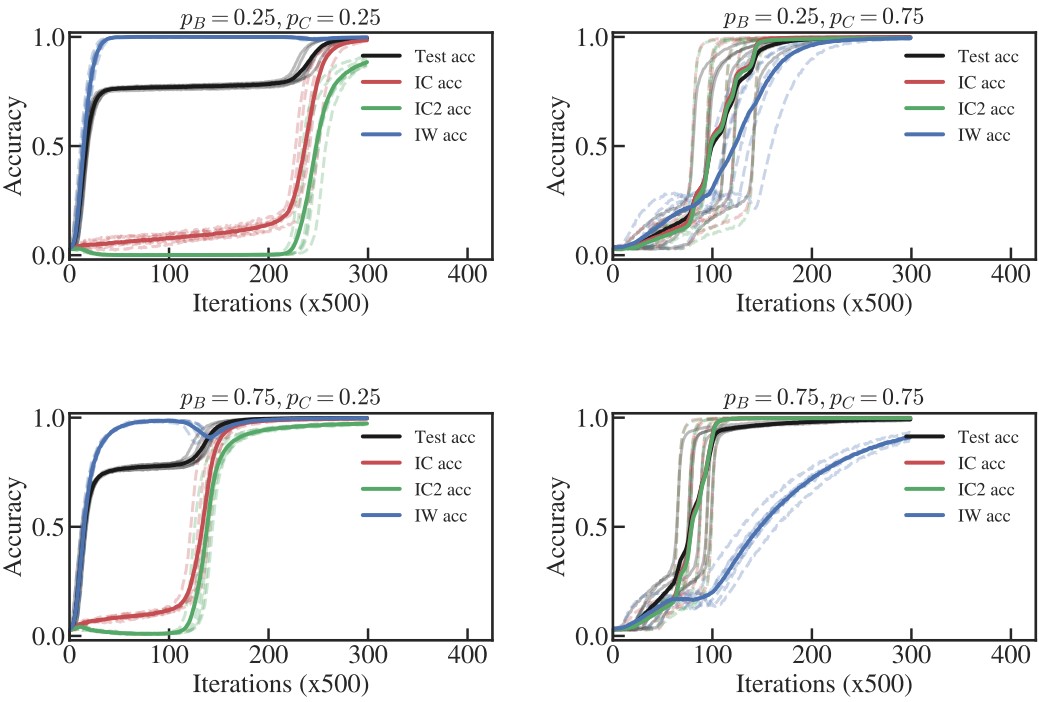

Figure A.1: Accuracy curves for the full model when $0 < p_B < 1$ and $0 < p_C < 1$. In all cases, the network learns both the ICL and the IWL solutions. Here $K = 256, B = 1, \alpha = 0, \varepsilon = 0$.

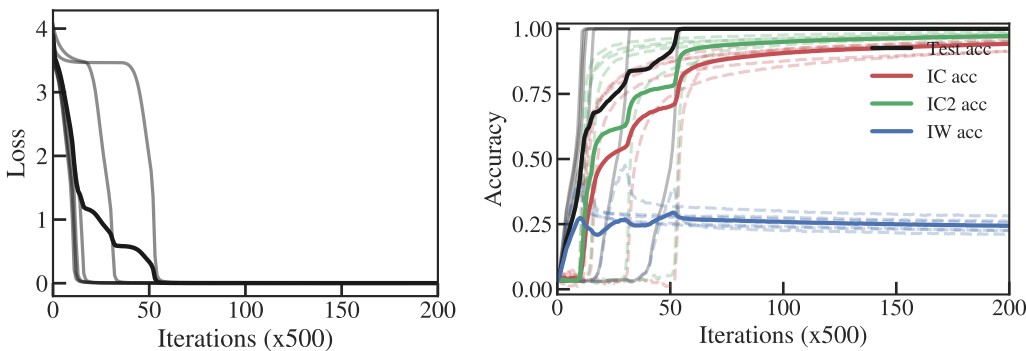

Figure A.2: Loss and accuracy curves for the minimal model. Here $K = 512, D = 64, B = 2$.

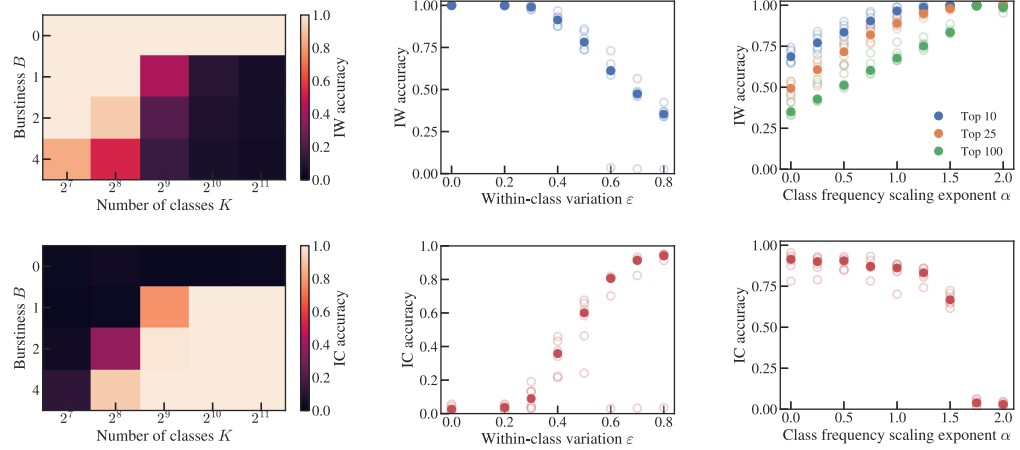

Figure A.3: Data distributional dependencies are recapitulated by the minimal model. Plotted as in Figure 2. Here $K = 512, D = 64, B = 1, \alpha = 0, \varepsilon = 0.1$ (except when that parameter is varied)

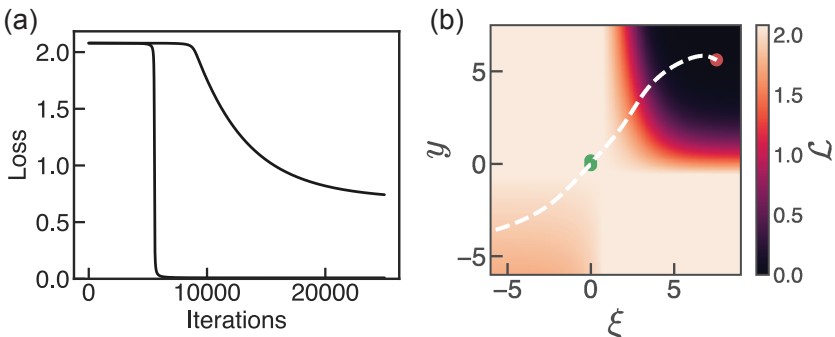

Figure A.4: (a) When $L = N$, the loss curves starting from two initial values recapitulate the two distinct behaviors noted in Figure 6c. (b) The loss landscape has a saddle at the origin such that small fluctuations lead the path either to the ICL solution (top right quadrant) or a sub-optimal minimum (bottom left quadrant).

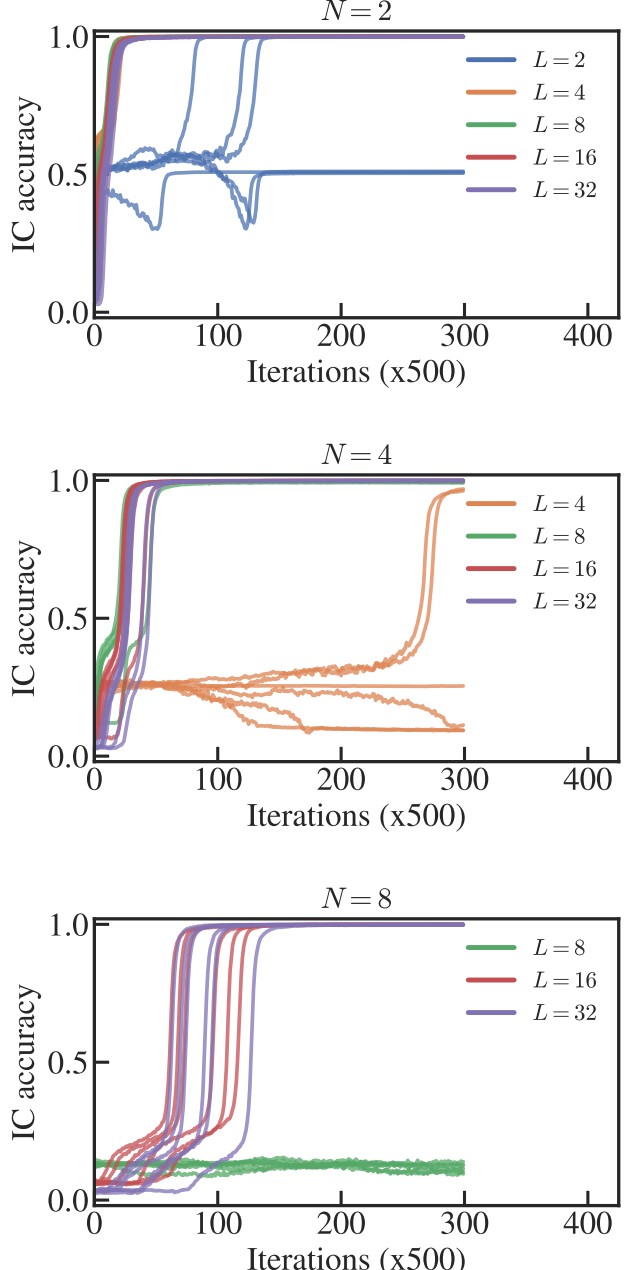

Figure A.5: IC accuracy curves for different $N$ and $L$ (six seeds for each pair of values of $L$ and $N$ are shown). Consistent with the theory and the minimal network, the full network (equation 2) robustly learns the in-context solution if $L > N$ but not when $L = N$. Here $K = 256, B = 1, p_C = 0.8, p_B = 1, \alpha = 0, \varepsilon = 0$.

