# OpenReview forum: "The mechanistic basis of data dependence and abrupt learning in an in-context classification task"
_ICLR.cc/2024/Conference — ICLR 2024 oral_

### Official Review · Reviewer_ciFG · 2023-10-31

**Soundness:** 2 fair
**Presentation:** 2 fair
**Contribution:** 3 good
**Rating:** 8
**Confidence:** 3

**Summary:**

This paper explores how and why properties of the data distribution control a transformer's propensity to use in-context vs. in-weights learning strategies.  The authors design a synthetic, parameterized family of tasks that expose dependencies of learning strategies on the data distribution that have been identified in prior work.  They then analyze the behavior of models trained on these tasks to understand how the dynamics of learning give rise to different strategies over time.  One of the key findings is that models learn a shortcut strategy that enables better-than-chance (but suboptimal) performance -- choosing an answer among one the labels presented in-context, without regard for the inputs presented in context.  In a stripped-down model that maintains the qualitative behavior of the full transformer parameterization, the authors show how this shortcut strategy facilitates (but is not necessary for) learning of a true ICL strategy via an analysis of the loss landscape.

**Strengths:**

This paper tackles a timely and interesting topic, and contains several insights and useful contributions.

-- The synthetic task family introduced is a clean and intuitive way of exposing dependencies of learning strategies on pretraining data distributions.  Showing that key phenomena identified in (Chan et al., 2022) can be replicated in this setting is a useful contribution.

-- The characterization of the initial slow learning phase as driven by an increase in context-label accuracy is interesting

-- The idea that a strategy that results in good context-label accuracy can facilitate (despite not being necessary for) learning of a true ICL strategy is very interesting, and some evidence is provided for this idea

-- The evidence provided that the emergence of induction heads is strongly linked with the development of the ICL strategy, while not entirely novel, is nice to see

**Weaknesses:**

I have the following concerns about this paper.  Many of them involve claims that I feel are made too strongly in the paper relative to the level of evidence provided.

-- The paper illustrates a set of phenomena in figure 2, and promises a mechanistic understanding of these phenomena.  But the mechanistic analysis provided later in the paper does not speak to most of the phenomenology -- for instance, the dependence of the ICL/IWL tradeoff on B, epsilon, K, and alpha.  In fact, the mechanistic analysis focuses on the p_C > 0 case, which is different from the p_c = 0 regime that gives rise to all the tradeoffs observed in Figure 2.  Thus, the connection between pages 1-4 of the paper and the rest is not entirely clear.

-- -- The following sentence, while intuitively reasonable, is written as a key strong claim and as far as I can tell is not really justified with evidence: "Therefore, the relative rates at which the network acquires ICL and IWL control the fraction of loss explained by each mechanism after convergence."

-- The paper makes strong causal claims based only on correlational evidence.  For instance, "Induction head formation drives the abrupt transition during ICL."  As far as I can tell no evidence is given for this claim, other than the (very suggestive, I agree!) fact that they coincide in time.

-- The paper makes strong claims about the three-parameter model proving or ruling out certain hypotheses.  An example is the sentence "This rules out hypothesis V as only the factors corresponding to the progress measures (a) through (d) have been included in the minimal model."  In my opinion, such claims are much too strong.  The three-parameter model is ultimately a different model from the original transformer architecture being used!  While the analysis of its behavior is suggestive of the learning strategies used by the original architecture, it is not conclusive.  The strength of the claims should be adjusted accordingly.

**Questions:**

-- In my opinion, the connection between the phenomenology observed in the full transformer model and the insights the authors derive from the three-parameter model could be made stronger with more experiments.  Have the authors considered techniques like activation/path patching and knockouts (see e.g. https://openreview.net/forum?id=NpsVSN6o4ul for examples of this approach) to test whether the explanations they come up with actually have explanatory power in the original model?

-- Why do the authors switch from the p_C = 0 case to the p_C > 0 case halfway through the paper?  I find this confusing as it makes the relevance of the mechanistic analysis to the p_C = 0 regime unclear.

-- The authors note that when p_B < 1, the network learns an IWL solution, and then fix p_B = 1 thereafter.  Later, they note that p_C > 0 always leads to an ICL solution (presumably with p_B fixed at 1).  To me, the authors have failed to consider the most realistic and interesting regime, where 0 < p_B < 1, and 0 < p_C < 1, where neither an ICL solution alone nor an IWL solution alone is optimal.  Is there a reason the authors choose not to consider this regime?

---

> ### Author Response · Authors · 2023-11-22
> **Response to Reviewer ciFG**
>
> Thank you for the insightful comments. We have made the following changes to address your concerns:
>
> -- The three-parameter model of the induction head makes a prediction that training is not robust when L = N due to the saddle point in the loss landscape at initialization. We now verify this in the full model with new experiments by exploring various values of L and N (see Figure A.5). The key observation is that, when L = N, either the network never learns the ICL solution (N = 8) or only learns for a few seeds (N = 4 and N = 2).
>
> Our simplistic two-layer/one head per layer model is not robust and will not learn the in-context solution after applying knockouts of the type considered in the suggested reference. However, note that our analysis methodology of constructing a minimal model of an induction head is somewhat similar to a "knockout". That is, we eliminate all the irrelevant aspects of the model and consider only the parts that implement the relevant operation. The reduced model reproduces most (if not all) of the phenomena shown by the full model. Together with the successful prediction of the outcome for L = N, we believe that this is rather strong evidence that the minimal induction head models do indeed capture the core computations performed by the full model.
>
> -- We use p_C > 0 to ensure that the network robustly and quickly learns the ICL solution. This was simply so that we could run each experiment for a shorter period and run more experiments overall. We now clarify this in the text. Note that we focus on ICL dynamics and not the ICL vs IWL tradeoff in this section and subsequent ones.
>
> We treat the quantification of the ICL vs IWL tradeoff separately from the analysis of the ICL learning dynamics. This is because a simple model can explain all of the ICL vs IWL results: the IC and IW solutions are essentially learned independently and at different rates that depend on the various hyperparameters. The tradeoff appears because there is a finite loss to explain. The relative rates of ICL and IWL determine the final IC and IW accuracies. The IC and IW accuracy curves do not show any evidence for a more complex interaction between IWL and ICL. We have modified the text to clarify this argument. We also adjust the strength of our claims to highlight that this is really a model that can explain the experimental data and not a conclusive proof.
>
>  -- Thanks for the nice suggestion, we had not considered the regime of 0 < pB < 1, 0 < pC < 1. We performed new experiments in this parameter range (see Figure A.1 of the revised paper). Note that in this regime, the network has to learn both the IC solution and the IW solution to achieve zero loss. Interestingly, we find that the network can learn and maintain both solutions simultaneously. This is again consistent with the picture that ICL and IWL are learned independently.

---

### Official Review · Reviewer_3R5R · 2023-11-01

**Soundness:** 4 excellent
**Presentation:** 4 excellent
**Contribution:** 3 good
**Rating:** 8
**Confidence:** 4

**Summary:**

Expanding on the seminal contributions of Chan et al and Olsson et al, this paper investigates the hugely important topic of the emergence of in-context learning in supervised learning via transformers. Specifically, it starts with experiments about the concurrence of in-context (IC) and in-weight (IW) learning, in the spirit of Chan et al, but systematically deconstructs their findings in such a way as to expose several potential hypotheses for the mechanism responsible for the ICL transition. The paper's main contribution is then to replicate and explain empirical phenomena via a three-parameter nested logits ansatz (based off a two-layer attention network and a linear classifier), with the appearance of an induction head controlled by a main parameter \xi seen as the difference between overlaps in on-diagonal and off-diagonal dot products feeding the final softmax.

**Strengths:**

The paper is excellently written and fairly easy to follow despite the depth of insights proven. The scientific investigation is very well conducted : of note is that it alternates particularly well between empirical elements, formulating subsequent hypotheses (section "Induction head formation drives the abrupt transition during ICL"), disproving some of those, and finally introducing a theory that accounts for those findings, replicating empirical stylized facts, whilst much simplifying the problem. In particular, the phenomenological model Equation 10 (and its illustration Figure 7) is a standout novel contribution, and clearly worthy of publication, in our view.

**Weaknesses:**

In a sense, the paper is tantalizing, as it invites further work, for instance on the interplay of overlap difference \xi and data Zipfianity parameter \alpha.

**Questions:**

What are the authors' intuition as to why does \xi undergo a phase transition driving ICL ? And how to, ideally, accelerate it using insights derived here ?

---

> ### Author Response · Authors · 2023-11-22
> **Response to Reviewer 3R5R**
>
> Thank you for the kind comments. Our model suggests that \xi changes abruptly due to the sequence of three nested logits that make up an induction head (see the Abrupt transitions during ICL section of the Discussion). We can potentially accelerate learning of \xi (or similar metrics for other tasks) by using a curriculum. We have added a few sentences in the discussion about the use of curricula in the Implications for LLMs section.

---

> > ### Comment · Reviewer_3R5R · 2023-11-22
> >
> > Thank you for your reply, and for sharing this work. I agree with the overall comments of my co-reviewers - and still believe that the phenomenological model is a standout contribution worthy of acceptance and highlighting at the conference. As such I will maintain my rating.

---

### Official Review · Reviewer_Y7Wm · 2023-11-02

**Soundness:** 4 excellent
**Presentation:** 3 good
**Contribution:** 4 excellent
**Rating:** 10
**Confidence:** 4

**Summary:**

The paper attempts to provide a deeper understanding of in-context learning of LLMs -- which is also related to the broader discussion about their "emergent learning capabilities".
To provide such an understanding, the authors wisely choose to abstract away many details of LLMs, and of the tasks that LLMs usually perform, and to focus instead on a simple experimental setting in which they can easily control or monitor whether the task is performed through in-weights learning (IWL) versus in-context learning (ICL). Additionally, the data is generated through a parsimonious gaussian mixture model in which they can also control some important aspects, such as the burstiness with which certain classes appear in the sequence, or the rank-frequency relation.
Then, and based on the insights from the previous experiments, they design a very small (in terms of number of parameters) phenomenological model of an "induction head" that reproduces quite well the observed behaviors of the more complex attention-based networks used in the experiments.

**Strengths:**

I have read several papers recently that try to provide some insights about the emergent capabilities of LLMs -- through abstract modeling and experiments with tasks such as linear regression (learned through ICL). This paper is the best I have read so far in that direction.
The experiments are wisely designed, allowing us to understand the complex tradeoff between IWL and ICL -- as well the effect of some key data distributional parameters such as the rank-frequency exponent.
The simple model proposed in the second part of the paper is also intriguing, explaining how the abstract model of an induction head can explain mechanistically the ICL capabilities of an attention-based network.

**Weaknesses:**

The paper can be improved in terms of writing/presentation. For example, you can explain early on in the paper what "induction head" means for readers that are less familiar with this area.

There are also several other parts of the paper in which the writing can be improved -- mostly by writing simpler/shorter/more clear sentences.

**Questions:**

As written in the Strengths section, I am very positive about this paper and so I do not have many technical suggestions or questions for the authors.

I would like to see at the end a clear discussion about the limitations of this simple/abstract model. Which aspects of an LLM's behavior may still be important but not captured by the proposed simple task and model that the paper proposes?

I would also like to see a more clear discussion of how your observations/conclusions agree (or disagree?) with earlier results in the literature about ICL and the emergent abilities of LLMs.

---

> ### Comment · Reviewer_Y7Wm · 2023-11-22
> **Can the authors please respond?**
>
> Even though my evaluation of this paper remains positive, I would very much like to see the authors' response to the thoughtful comments of reviewer ciFG.
>
> Thank you.

---

> ### Author Response · Authors · 2023-11-22
> **Response to Reviewer Y7Wm**
>
> Thank you for the kind comments. The discussion has been significantly expanded to the extent that we could given the space limitations. We have added subsections briefly expanding on limitations, relationship with past work and implications for LLMs. The text has been revised for clarity based on your comment and reviewer sq8T's detailed suggestions.

---

### Official Review · Reviewer_sq8T · 2023-11-08

**Soundness:** 4 excellent
**Presentation:** 3 good
**Contribution:** 3 good
**Rating:** 10
**Confidence:** 4

**Summary:**

The authors build on recent work on induction heads as the mechanism of in context learning (ICL) and give a characterization of how they are learned.  They start with a minimal transformer architecture, show that it capture previous findings on data dependence of ICL vs in weights learning (IWL) and proceed to carry our further model reductions that focus in on crucial abstractions that characterize the learning dynamics in terms of abstract underlying variables rather than particulars of particular connection weights.  They test several causal hypotheses, zeroing in on factors that jointly influence how learning occurs.

**Strengths:**

The ability to use information in context to respond appropriately to later queries (called 'in context learning' or ICL), is central to the capabilities of AI systems like ChatGPT.  ICL was enabled by the attention mechanism in transformer-based neural networks.  ICL is exemplified by the simple item-label association task (introduced by others) that the authors have selected for the focus of their analysis.  By shedding light on how this task is solved (building on an earlier paper taking initial steps in this direction) the current paper deepens our understanding of this core property of today's performant AI systems.

The authors have created new variants of the task that further simplify it, and have introduced minimally-sufficient transformer architecture containing two attention layers, which together implement what they call an 'induction head', arguably the core emergent computational structure enabling ICL.  They have provided a insightful analysis of the (ultimately simple, but nevertheless important) computations performed by the network that allow the effective use of information in context in their task.  They gone on to attempt to understand how this attention head computation emerges as the network learns to solve the item-label association task.  They identify progress measures in both the network's input-output performance and of its attention head computations and establish clear alignments of several of these measures.  I consider these measures and their alignments enlightening contributions and consider them to be strengths of the paper.

They go on to further support their analysis by developing a three-parameter reduction of the induction head, and show that the learning dynamics of this reduction is sufficient to reproduce many of the features of the learning dynamics of their complete neural network; they then use the reduction to test hypotheses about the relationships between the progress measures, showing that a further reduction that eliminates one of the progress measures makes learning success initialization dependent.  Finally, they make an even further reduction in the form of a 2- or 3- parameter 'phenomenological model' whose loss landscape can be fully characterized.  The parameters now directly reflect the efficacies of the two attention layers making up the induction head and of their mapping to the correct label, and allow the loss landscape of each of the variants to be visualized.  This phenomenological model provides an abstract characterization of the emergent learning dynamics of the 2- and 3 parameter reduction models that allows a full explanation of why these models learn reliably under the condition that the number of possible labels is greater than the number of item-label pairs in context, and fails to learn reliably when the number of possible labels is equal to the number of item-label pairs in context.  All the paper provides us with important clues toward understanding the computations performed by transformers and of the processes that give rise to their learning dynamics.  Along the way the paper provides an approach to analysis of neural network learning dynamics that others could adopt to understand the learning dynamics as they arise in other setting, another valuable contribution to the effort to understand the complex computations performed by neural networks.

**Weaknesses:**

Although I consider the analysis presented a tour de force, possessing all the strengths describe above, it is not perfectly clear that the analysis of the 2-3 parameter reduction would carry over to the full 2-attention-head network of Figure 1c.  A hunch I have is that the L=N case might not be quite as susceptible to failure in the full network because the full network might have a more complex loss landscape with a lower likelihood of being initialized in a place in that landscape that doesn't allow a complete solution.  An important and simple step toward addressing this would be to repeat the L = N simulation in the full network.  If the full network fails to learn in that case, it would confirm the applicability of the analysis to the full network.  Success would not fully invalidate the analysis, but would leave something left to explain.

More generally, I believe more consideration of what will happen in a larger model will be useful for the field.  Clearly things will not work just in the way they do in these reductions when the task is learned in a larger transformer.  While fuller characterization of that will be a task for future work, noting this issue as a limitation of the present effort and pointing considering how these results inform us about what is happening in LLMs will be valuable.

There are two less important weaknesses I'd like to see addressed.

First, I don't feel I have an intuitive understanding of why the loss landscape of the 3 parameter model does not have a saddle point at the point were all three parameters are equal to 0.  Perhaps an understanding of this is latent in the equations and I could work it out with a bit of effort, but to help me (and possibly others) understand, it would be useful if the authors could work out such an intuitive understanding.  Such an understanding could help address reasons why the behaviors of the 2- and 3-parameter reduced models might or might not be applicable to the full model.

Second, paper is harder to read than it should be.  The main deficiency of the paper was its failure to take cognizance of the difficulty of extended chains of arbitrary associative bindings requiring long-distance leaps across context.  It is just such binding that lie at the heart of the mechanisms the authors are investigating, but they are hard for human readers when arbitrary as they often were in this paper.

As examples, we are treated to terms like the former vs the latter as referring expressions, arbitrary labels (a-d) for key phenomena, random ordering of the assignments of these labels to lines in graphs, arbitrary labels for hypotheses (I-V), and the unhelpful placement of figures (esp fig 4) on pages remote from the place in the paper where they are discussed.  Although ultimately the conclusions are stated in (what I find myself to be) conceptual terms, there should be engagement with this conceptual structure in the referential expressions used. I know space is limited, but I'm sure it is possible to do a better job. As examples, H3 could be abbreviated sCLA -> ILA+TILA (slow-learned context-label attention -> Item-label attention and Target-item-label association). Just let a,b,c,d and I-V go.  H4 and H5 should each be expressed directly, or at the very least the order of defining the symbols x and 0/ should correspond to their order of appearance in these hypotheses.

I am also not sure that the difficulty of the L=N case in the

**Questions:**

Suggestion 1: Confirm that they setting L=N and B=1 disrupts learning of the induction head in the full model as it does in the 2/3 parameter reduction and in the corresponding phenomenological model.

Suggestion 2: Redirect some of the presentation to an appendix to provide at least 1/3-1/2 page for discussion of implications for LLMs.

Suggestion 3:  Provide an intuitive understanding of the shape of the loss landscape in both the 2 and 3 parameter version of the phenomenological model.

Suggestion 4: Reduce the cognitive load on the reader: Specifically, increase the conceptual content of referential expressions throughout.  Don't use arbitrary letters/roman numerals for quantities and hypotheses; avoid former/latter and 'respectively' type constructions where possible.  Also, place figures as close as possible to the point in the text where they are described.

---

> ### Comment · Reviewer_sq8T · 2023-11-21
> **I have been expecting some response and revision from the authors of this ms -- rating could go down without it**
>
> Although I still think this paper reports an important set of results and contains deep analyses, it lacks accessibility.  Another reviewer commented on the sufficiency of the reduction and therefore at least an acknowledgement of this is required.  Addressing these two concerns in a revision will be the minimum I require to retain my rating of 10 for this ms

---

> ### Author Response · Authors · 2023-11-22
> **Response to reviewer sq8T**
>
> Thank you for the kind comments and valuable suggestions. Here is a summary of the changes we have made in response to the suggestions:
>
> 1. New experiments confirm that setting L = N, B = 1 in the full model disrupts learning. Please see Figure A.5 in the new version. None of the six seeds learn the ICL solution when N = 8 and L = N, though some of the seeds do learn for smaller context windows. It's possible that using a different optimizer or scaling the initial query/key/value matrices may lead to different results. However, this will not change the interpretation that the loss landscape at initialization is flat when L = N.
>
> 2. Figure 8 has been moved to the Appendix (now Figure A.4) to provide more space for an expanded discussion. We contextualize our work, comment on implications for LLMs and briefly discuss limitations.
>
> 3. A few sentences have been added to provide some intuition (last two paragraphs before the discussion). Note that the origin is not a critical point (and thus not a saddle) of the loss except when L = N. This is because the network gradually aligns the regression vectors with their corresponding labels (increasing $\xi$) when learning to randomly pick one of the contextual labels in the slow phase.
>
> 4. We have removed the arbitrary labels for progress measures and hypotheses, shrunk/split long sentences and moved the figures. The text has been revised for clarity. The current placement of the figures may still not be ideal (esp fig 4), but the page limit makes it hard to move figure 4 around.

---

> > ### Comment · Reviewer_sq8T · 2023-11-23
> > **Improved paper - still a 10, with a comment on the value of isolating IWL after considering ICL and IWL together**
> >
> > The authors have now added useful results showing that the predictions of the reduced models hold up in the actual (albeit minimal) transformer, supporting the relevance of these models to that case.  The cleaner writing is also an enhancement, as are the new results with in the 0 < pB < 1, 0 < pC < 1 regime, suggested by reviewer ciFG.
> >
> > Reviewer ciFG questioned 'the connection between pages 1-4 of the paper and the rest', that is, first replicating the phenomenology from Chan et al on the interplay of ICL and IWL, then beaming in of the ICL / induction head by setting pB=0 and pC=1.  I consider this progression a plus.  The fact that the paper's minimal induction head transformer with two attention heads and one FF network replicates the Chan et al findings makes it clear that it can capture the mix of IW and ICL generally present in transformers, making it a useful reduced model to further understand. Zeroing in to understand more about ICL in an architecture that can capture both is important because IWL is the (or at least one of the) key new emergent properties that transformers provide, enabling few-shot learning, a crucial attribute previous models lacked.  Lots for further work will grow from this, including more work to understand such things as whether the analysis presented here still applies in the mixed IW/IC setting; it might still apply to the emergence of ICL, given that the IW learning probably takes place completely through the direct weights in the transformer FF layers, bypassing the attention heads which are especially relevant for ICL.  Perhaps an added comment about this in the discussion could address ciFG's concern, and further enhance the appreciation of the importance of the paper.

---

### Meta-Review · Area_Chair_xcbd · 2023-12-09

**Metareview:**

All reviewers felt this was an extremely interesting and illuminating paper that explored and characterized in a hypothesis-driven way the emergence of in-context learning in transformers. Reviewers especially appreciated the combination of theoretical foundation-laying and empirical experiments specifically designed to test the assumptions of those theories. They clearly found the results fascinating and illuminating, and all strongly supported acceptance.

The main weaknesses identified were clarification questions posed by ciFG, which were subsequently answered in the authors’ rebuttal, as well as requests for a clearer presentation of the work, which I have no doubt could be easily addressed in the camera-ready.

This paper should probably be highlighted at the conference, as it's likely to draw a lot of interest.

**Justification For Why Not Higher Score:**

NA

**Justification For Why Not Lower Score:**

It should be presented at the conference in the form of a talk, and probably an oral as it's likely to drawn widespread interest.

---

### Decision · Program_Chairs · 2024-01-16

Accept (oral)